# Biofertilization with PGP Bacteria Improve Strawberry Plant Performance under Sub-Optimum Phosphorus Fertilization



Pedro Valle-Romero [1], Jesús V. García-López [2], Susana Redondo-Gómez [1], Noris J. Flores-Duarte [3], Ignacio D. Rodríguez-Llorente [3], Yanina Lorena Idaszkin [4,5], Eloisa Pajuelo [3] and Enrique Mateos-Naranjo [1,*]

[1] Departamento de Biología Vegetal y Ecología, Facultad de Biología, Universidad de Sevilla, 41012 Seville, Spain
[2] Servicio General de Invernadero, Centro de Investigación, Tecnología e Innovación (CITIUS), Universidad de Sevilla, 41012 Seville, Spain
[3] Departamento de Microbiología y Parasitología, Facultad de Farmacia, Universidad de Sevilla, 41012 Seville, Spain
[4] Instituto Patagónico para el Estudio de los Ecosistemas Continentales (IPEEC-CONICET), Boulevard Brown, 2915, Puerto Madryn 9120, Argentina
[5] Departamento de Química, Facultad de Ciencias Naturales y Ciencias de la Salud, Universidad Nacional de la Patagonia San Juan Bosco, Boulevard Brown, 3051, Puerto Madryn 9120, Argentina
* Correspondence: emana@us.es

**Abstract:** Biofertilization with plant growth-promoting bacteria (PGPB) could optimize chemical fertilization for strawberry crop cultivation. A greenhouse study was arranged to assess the impact of an isolated PGPB consortium from halophytes on strawberry development, physiological traits, and nutritional balance subjected to two phosphorus fertilization limitation treatments (with and without insoluble phosphorus form application). Biofertilization had a positive effect on strawberry development. Thus, shoot and root biomass was c. 20 and 32% higher in inoculated plants grown with insoluble phosphorus. This effect was mediated by a positive bacterial impact on plant carbon absorption capacity and water use efficiency, through a reduction in $CO_2$ diffusional and biochemical photosynthesis limitation. Thus, net photosynthetic rate and intrinsic water use efficiency showed increments of 21–56% and 14–37%, respectively. In addition, inoculation led to a better efficiency of the plant photochemical apparatus, as indicated by the invariable higher PSII photochemistry parameters. Furthermore, these effects correlated with improved nutritional balance of phosphorus and nitrogen, which was directly related to the beneficial impact on carbon metabolism and, consequently, on strawberries' growth. In conclusion, we can recommend the biofertilization based on PGPB for achieving more efficient strawberry P fertilization management practices, providing high efficiency in yields.

**Keywords:** biofertilizers; chlorophyll fluorescence; gas exchange; nutrient deficit; PGPR; phosphorous; strawberry

## 1. Introduction

Strawberry cultivation has undergone remarkable development in the last two decades around the world due to its high yield and economical cost effectiveness, with worldwide production greater than 7.7 million tons per year [1], especially under integrated intensive protected environment crop systems inside plastic greenhouses using drip irrigation and fertilization systems [2].

Despite the socioeconomic importance of this crop, its intensity character and the need to provide larger amounts of inputs have been linked to environmental issues such as surface and underground water pollution by residues of chemical fertilizers [3,4], circumstances that jeopardize its maintenance and sustainability in the short- and medium-term,

since consumers require the production of high-quality foods and products able to satisfy their demand, while minimizing negative impacts on the environment [5]. Therefore, one of the main goals for the strawberry industry is to encourage research aimed at optimizing the fertilization process, minimizing the use of industrial fertilizers, and thereby reducing costs of production and preventing environmental risks.

Intensive work has been done to get to know the most adequate management of strawberry crop fertilization. Therefore, the studies have focused on: the influence of different doses of industrial fertilizers on vegetative development and strawberry yield [6,7], the effect of chemical and organic fertilizers [7,8], the use of organic fertilizers as opposed to chemical fertilizers before planting and during the development of the crop [9,10], and the combination of chemical and biofertilizers [11,12]. However, despite this progress, strawberry cultivation still uses a large number of agrochemicals, which contribute to maintaining high production levels but entail high economic costs for producers, not to mention the aforementioned obvious environmental risk. In this way, positive rhizospheric interactions due to microbiota naturally present in the soil or supplied in the form of biofertilizers positively affect soil quality [13–15] and can be exploited as an alternative to agrochemicals to improve crop yield and reduce chemical input [16–18].

Among the potential beneficial soil microorganisms of plants, bacteria with plant growth-promoting properties (PGPB) have demonstrated a variety of beneficial effects on plant performance and adaptability against different environmental factors, including nutrient deficit [19–24]. Therefore, mechanisms such as phosphorous solubilization, siderophores production with improved iron uptake, and nitrogen fixation are the main beneficial properties of PGPB that could be involved in more efficient plant fertilization management practices, providing high cost-effective yields [25]. Additionally, PGPB can produce phytohormones such as auxins, stimulating root elongation and improving plants' nutrient acquisition [26,27]. The study of the potential of PGPB to improve the efficiency of strawberry fertilization is in its infancy, as in general most of the work on strawberry and microorganisms has focused primarily on the potential of bacteria as agents for the biocontrol of soil pathogenic fungi [28,29]. Therefore, a challenge exists in determining PGPB potential to improve strawberry development, in a scenario of fertilizer input restriction which would be useful for identifying the role played by bacteria in improving the response of strawberry plants in nutrient-poor soils. Taking into account this aspect, we hypothesize that the potential negative effect of a limiting phosphorus fertilization process on strawberry responses could change due to the presence of a bacterial consortium integrated by bacterial strains with multiple PGP properties that would exert an effect on the activation of concrete plant tolerance mechanisms to nutrient-deficit stress favoring its nutritional balance and, consequently, its growth and physiological performance.

To explore this, we used a full factorial greenhouse study approximation to shed light on the influence of a PGP bacterial consortium on strawberry plant (*Fragaria vesca* var. *Rociera*) growth, photosynthetic apparatus performance, and ion homeostasis under different phosphorus fertilization treatments (fertilization without phosphorus and with an insoluble form of phosphorus).

## 2. Material and Methods

### 2.1. Experimental Setup

Strawberry plants (*Fragaria vesca* var. *Rociera*) with a well-developed crown were seeded in 2.5 L plastic pots filled with sterilized commercial vermiculite substrate ($SiO_2$ 34–43%, $Al_2O_3$ 7–15%, $Fe_2O_3$ 5–13%, $MgO$ 20–28%, $CaO$ 0.2–1%, $TiO_2$ 0.01–0.1%, and 0.5–3 mm grain size; Massó Garden SA) and kept in a greenhouse (University of Sevilla Research, Technology and Innovation Centre, CITIUS II; 37°24′ N, 6°0′ W; Southwest Spain) with 21–25 °C, 40–50% RH, and subjected to a day/night regime of 16 h of light (maximum photosynthetic photon flux density (PPFD) incident on leaves of 1000 µmol m$^{-2}$ s$^{-1}$) and 8 h of darkness and adequate irrigation and fertilization until the beginning of the experiment.

In February 2022, after two months of growth of plants under conditions described previously, fully developed plants were divided into two biofertilization treatment groups (inoculated and without bacterial application) in combination with two treatments for the availability of phosphorus fertilization (with and without phosphorus; $n = 40$, 10 plants per treatment). Therefore, the following treatments were obtained: (i) non-inoculated and fertilization with BNM medium [30] without phosphorus, Non-Inoc. + $P^-$; (ii) non-inoculated and fertilization with BNM medium supplemented with an insoluble phosphorus form $Ca_3(PO_4)_2$, Non-Inoc. + $P^+$; (iii) bacterial inoculation and fertilization with BNM medium without phosphorus, Inoc. + $P^-$; and (iv) bacterial inoculation and fertilization with BNM medium with insoluble phosphorus, Inoc. + $P^+$. The experiment was developed for four months under the aforementioned environmental conditions, and during this time, fertilization treatments and irrigation were applied twice a week with a separation of 3 days (applying 100 mL per plant). Additionally, the bacterial inoculation was carried out biweekly, applying 100 mL of a previously designed bacterial inoculum.

### 2.2. Bacteria Strain Selection, Characteristics and Inoculum Preparation

The bacterial inoculum used was composed of five bacteria strains, which had been isolated from several halophyte rhizospheres: *Spartina densiflora*, *Spartina maritima*, *Salicornia ramosissima*, and *Halimione portulacoides* and from the Belgian Co-ordinated Collection of Microorganisms (BCCM). The strains were named: *Variovorax paradoxus* (S110), *Pseudomonas* sp. (SDT3), *Bacillus velezensis* (SMT38), *Pseudarthrobacter oxydans* (SRT15), and *Bacillus zhangzhouensis* (HPJ40); for the isolation, characterization, identification, and conservation details, see our previous studies [18,31–34]. Bacterial strains were selected because they had growth self-compatibility and exhibited high multi-stress resistance and a variety of complementary plant growth-promoting (PGP) properties, aspects that make them an ideal inoculum to be adapted to and employed in several agronomic management contexts and environmental conditions, thus being able to improve fertilization process efficiency in strawberry crops. Table 1 shows the PGP properties that were analyzed in several of our previous studies [18,31–34]. For the inoculum preparation technical details, see Navarro-Torre et al. [35] and Flores-Duarte et al. [36]

**Table 1.** Biofertilizer bacterial strains' PGP properties.

| PGP | SDT3 | HPJ40 | SMT38 | SRT15 | S110 |
|---|---|---|---|---|---|
| Phosphate solubilizing | + | + | - | + | - |
| Siderophores | + | + | + | - | + |
| IAA | - | - | - | + | + |
| Fixation of nitrogen | - | + | + | + | + |
| Biofilm | - | + | + | + | + |
| ACC activity | n.d | - | + | - | + |

SDT3: *Pseudomonas* sp, HPJ40: *B. zhangzhouensi*, SMT38: *B. velezensis*, SRT15: *P. oxydans*, and S110: *V. paradoxus*; +, positive; -, negative; n.d., non-determined.

### 2.3. Plant Growth and Physiological Traits Analysis

After 120 days of treatment, plants were harvested and their dry mass fractions were recorded. Additionally, the leaf water content of fully developed randomly selected leaves was determined ($n = 10$; for details, see Redondo-Gómez et al. [17]). Additionally, 40, 60, 80, 100, and 120 days after the application of experimental treatments, leaf instantaneous gas exchange and photochemical traits characteristics were recorded in fully developed leaves from each inoculation and phosphorus fertilization treatment combination ($n = 10$) using an infrared gas analyzer (LI-6800-01, LICOR Inc., Lincoln, NE, USA) and a fluorimeter (FMS-2; Hansatech Instruments Ltd., King's Lynn, UK), respectively. Thus, net photosynthetic rate ($A_N$), stomatal conductance ($g_s$), intercellular $CO_2$ concentration ($C_i$) and intrinsic water use efficiency ($_iWUE$), actual efficiency of the photosystem II ($\Phi_{PSII}$), and the electron transport rate (ETR) were measured at midday between 11:00 and 13:00 pm at 1000 µmol

photon m$^{-2}$ s$^{-1}$ (with 15% blue light), 2.0–3.0 kPa vapor pressure, $50 \pm 1\%$ air relative humidity, 400 μmol $CO_2$ mol$^{-1}$, and 25 °C. Finally, maximum quantum efficiency of PSII photochemistry ($F_v/F_m$) was obtained in dark-adapted leaves according to the protocol described by Mateos-Naranjo et al. [37].

### 2.4. Quantitative Analysis of Leaf Photosynthetic Limitations

After 120 days of phosphorus fertilization treatment and bacterial inoculation application, 5 $A_N/C_i$ curves with 12 different $C_a$ values per treatment combination were developed between 10:00 and 14:00 h in fully developed leaves, using an infrared gas analyzer (LI-6800-01, LICOR Inc., Lincoln, NE, USA) to determine the absolute limitations of the observed decrease in $A_N$ by stomatal conductance (SL), mesophyll conductance (MCL), and biochemistry (BL) according Grassi and Magnani's [38] approach. Thus, using the curve fitting method, the mesophyll conductance ($g_m$) and the maximum carboxylation rate allowed by ribulose-1,5-biphospate (RuBP) carboxylase/oxygenase ($V_{c,max}$) were obtained following the recommendations of Flexas et al. [39,40], Pons et al. [41], and Long and Bernachi [42], since this method required those parameters together with $A_N$ and $g_s$.

### 2.5. Ion Concentration in Leaves and Roots

Calcium (Ca), potassium (K), magnesium (Mg), manganese (Mn), and phosphorus (P) in strawberry plant dry samples were analyzed via ICP-OES spectroscopy (Thermo ICAP 6500 DUO, Waltham, MA, USA) after 120 d of treatment. In addition, undigested dry sample total nitrogen (N) and carbon (C) concentrations were recorded using an elemental analyzer (Leco CHNS-932, Madrid, Spain) ($n = 10$).

### 2.6. Statistical Analysis

Statistica software v. 10.0 was used for statistical analyses. Thus, the effect of phosphorus and bacteria inoculation treatments on strawberry plant development and physiological and nutrient homeostasis was determined using generalized linear models (GLM), followed by post hoc LSD test (i.e., Fisher's least significant difference) for multiple comparisons.

## 3. Results

### 3.1. Bacterial Inoculation and Phosphorus Fertilization Availability Effect on Strawberry Growth

Inoculated plants had a lower level of foliar damage level (i.e., chlorosis and necrosis) and a higher density of leaves and roots than their non-inoculated counterparts, with this effect being more evident in those fertilized with the form of insoluble phosphorus after 120 days of treatment. Thus, bacterial inoculation increased shoot and root weights by c. 20 and 32% in these plants (Figure 1A,B). Finally, there was also a significant effect of bacterial inoculation on LWC, so that the inoculated plants had higher LWC regardless of fertilization treatment (GLM, inoc., $p < 0.05$; Figure 1C).

### 3.2. Bacterial Inoculation and Phosphorus Fertilization Availability Effect on Strawberry Photosynthetic Apparatus Performance

In general, inoculated and fertilized plants with BNM medium supplemented with insoluble phosphorus form showed the highest $A_N$ values, followed by those inoculated but grown in the absence of phosphorus and non-biofertilized plants regardless of fertilization treatment (Figure 2A). Compared to inoculated and fertilized plants with phosphorus, the $A_N$ values decreased by 25, 34, and 37% in Inoc. + P$^-$, non-inoc. + P$^+$, and non-Inoc. + P$^-$ treatments, respectively, after 120 d (Figure 2A). A very similar pattern was observed for $g_s$; however, $g_s$ values did not vary between biofertilized plants grown in the absence of phosphorus fertilization and non-inoculated plants independent of fertilization treatment after 80, 100, and 120 days of treatment (Figure 2B). Furthermore, $C_i$ values tended to increase in non-inoculated plants for both fertilization treatments throughout the days of the experiment compared with their inoculated counterparts, although the lowest $C_i$ values were recorded in those fertilized with insoluble phosphorus after 80, 100, and 120 days of

experiment (GLM: Inoc., $p < 0.01$; Fert., $p < 0.01$; Figure 2C). Additionally, $_i$WUE values were higher after bacterial inoculation and fertilization with phosphorus after 80 days of treatment, and this difference was even greater after 100 and 120 days (Figure 2D).

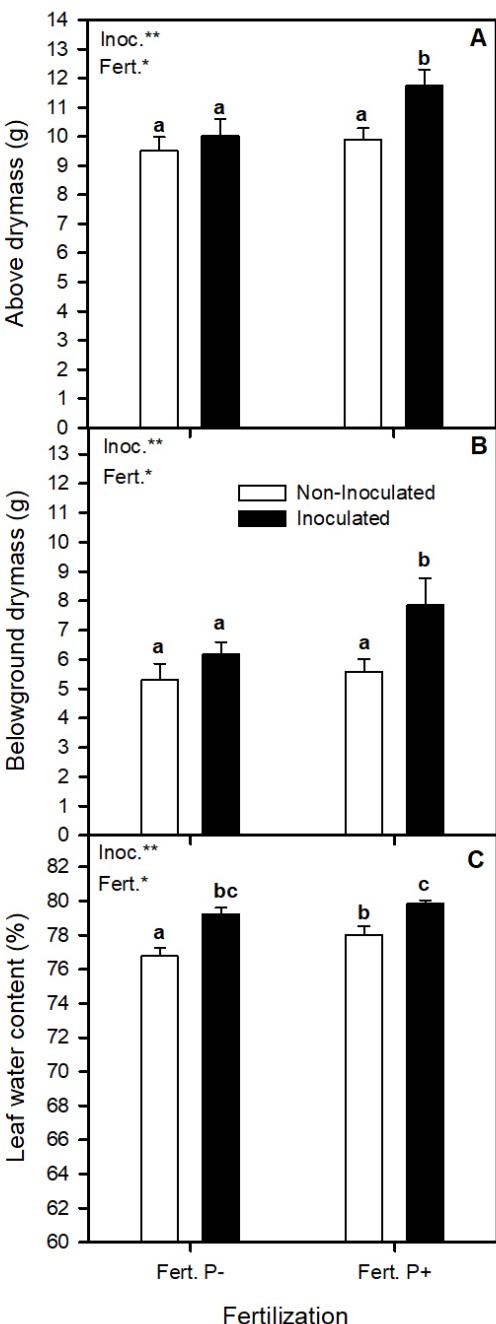

**Figure 1.** Aboveground dry mass (**A**), belowground dry mass (**B**), and leaf water content (**C**) in strawberry plants fertilized with BNM medium without phosphorus (Fert. P−) or with BNM medium supplemented with an insoluble phosphorus form (Fert. P+) and subjected to two inoculation treatments after 120 days of treatment. Values represent mean ± SE of 10 replicates. Inoc., Fert., or Inoc. x Fert. in the panel corner indicate principal or synergistic significant effects. Different letters indicate means that are significantly different from each other. (LSD test, * $p < 0.05$, ** $p < 0.01$).

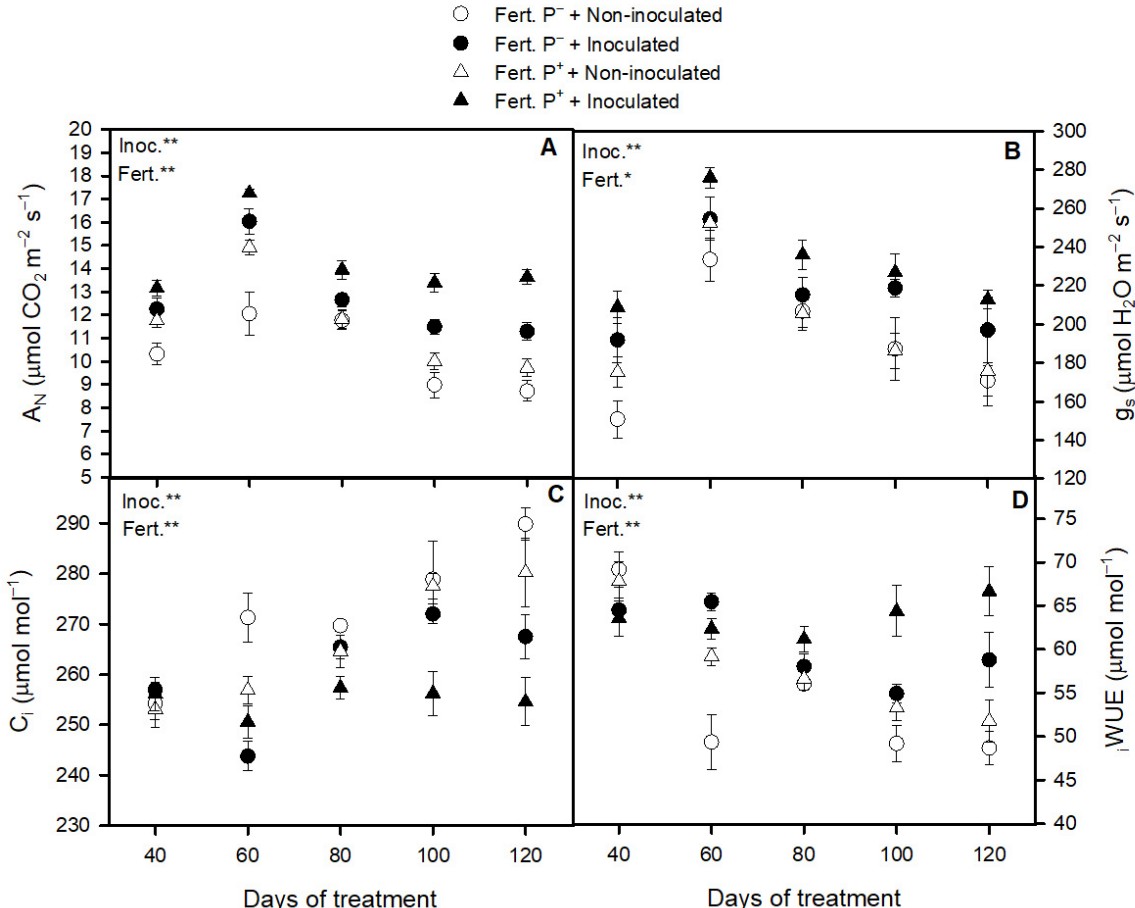

**Figure 2.** Net photosynthetic rate, $A_N$ (**A**), stomatal conductance, $g_s$ (**B**), intercellular $CO_2$ concentration, $C_i$ (**C**), and intrinsic water use efficiency, $_iWUE$ (D) in strawberry plants fertilized with BNM medium without phosphorus (Fert. P−) or with BNM medium supplemented with an insoluble phosphorus form (Fert. P+) and subjected to two inoculation treatments after 40, 60, 80, 100, and 120 days of treatment. Values represent mean ± SE of 10 replicates. For statistical details, see Figure 1 caption. (LSD test, * $p < 0.05$, ** $p < 0.01$).

However, the quantitative limitation analysis of photosynthesis indicated that the relative importance of diffusional and biochemical limitations varied with bacterial inoculation and its effect was modulated by phosphorus fertilization (Figure 3). Therefore, the reduction in $A_N$ in inoculated and fertilized plants without phosphorus supplementation was mainly due to diffusional limitations, with percentages of total photosynthetic inhibition of 20, 6, and 9% for MCL, SL, and BL, respectively. However, BL was dominant in the non-inoculated plants, with 27 and 23% of total limitations in plants fertilized without and with an insoluble phosphorus form, respectively (Figure 3).

These results were linked by $g_m$ and $V_{c,max}$ responses; thus, the highest $g_m$ values were recorded in plants that were inoculated and fertilized with insoluble phosphorus (GLM: Inoc. x Fert., $p < 0.01$; Figure 4A). Additionally, $V_{c,max}$ values showed a similar trend, although inoculated plants grown without phosphorus addition also showed significantly higher values than their non-biofertilized counterparts (GLM: Inoc., $p < 0.01$; Figure 4B).

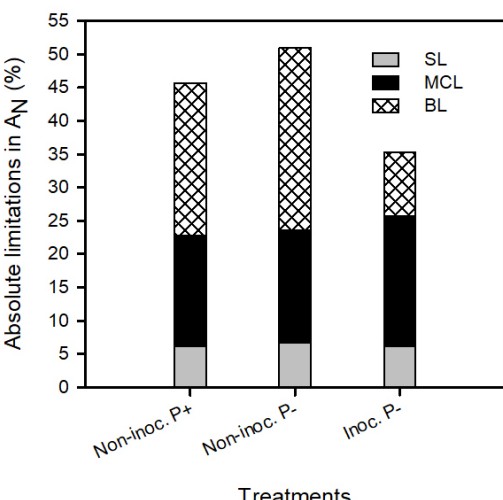

**Figure 3.** Quantitative analysis of absolute limitations in photosynthesis ($A_N$) of strawberry plants fertilized with BNM medium without phosphorus (P−) or with BNM medium supplemented with an insoluble phosphorus form (P+) and subjected to two inoculation treatments after 120 days. SL, stomatal; MCL, mesophyll; BL, biochemical limitations.

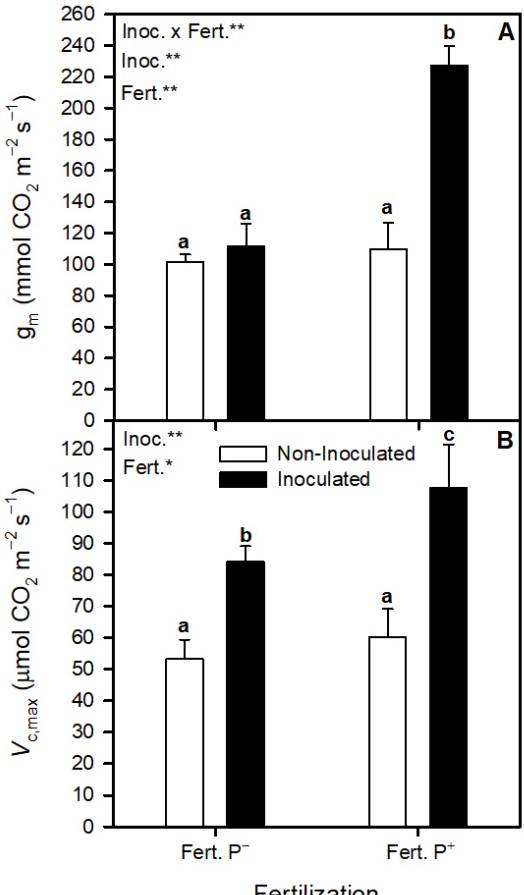

**Figure 4.** Mesophyll conductance, $g_m$ (**A**) and maximum carboxylation rate allowed by RuBP, $V_{c,max}$ (**B**) in strawberry plants fertilized with BNM medium without phosphorus (Fert. P−) or with BNM medium supplemented with an insoluble phosphorus form (Fert. P+) and subjected to two inoculation treatments after 120 days of treatment. Values represent mean ± SE of five replicates. For statistical details, see Figure 1 caption. Different letters indicate means that are significantly different from each other. (LSD test, * $p < 0.05$, ** $p < 0.01$).



### 3.3. Bacterial Inoculation and Phosphorus Fertilization Availability Effect on Strawberry Chlorophyll Fluorescence

$F_v/F_m$ values were higher in inoculated plants compared to their non-inoculated counterparts, especially in the absence of phosphorus supplementation after 80 and 120 days of treatment (i.e., Fert. P− treatment; GLM: Inoc., $p < 0.01$; Figure 5A). Furthermore, $\Phi_{PSII}$ and ETR values were invariably higher with bacterial inoculation and fertilization with insoluble phosphorus throughout all sampling periods (GLM: Inoc. x Fert., $p < 0.05$ and $p < 0.01$; Figure 5B,C). However, there was also a significant positive effect of a lesser magnitude for bacterial inoculation in the absence of phosphorus fertilization after 100 and 120 days (GLM: Inoc., $p < 0.01$; Figure 5B,C).

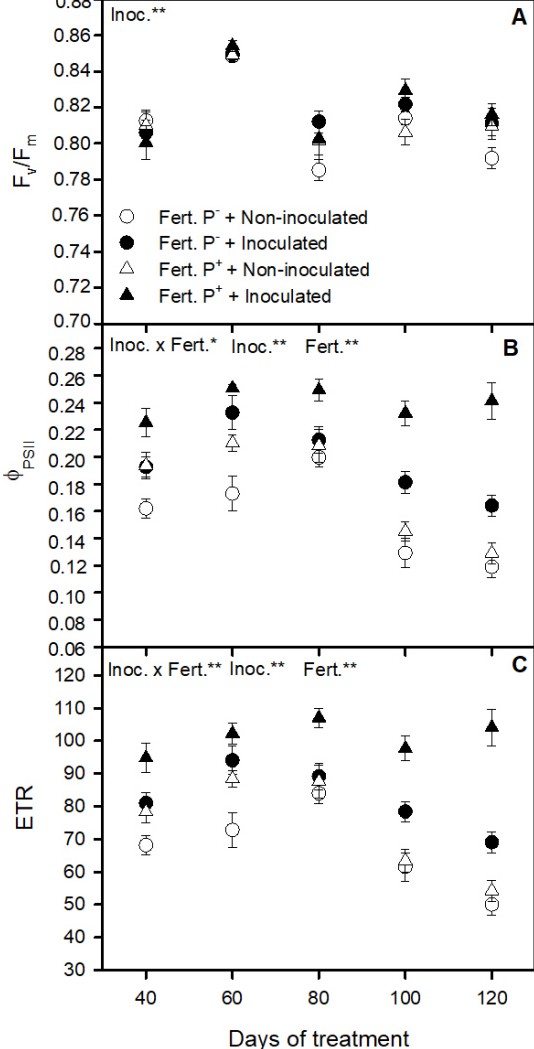

**Figure 5.** Maximum quantum efficiency of PSII photochemistry, $F_v/F_m$ (**A**), actual efficiency of photosystem II, $\Phi_{PSII}$ (**B**), and the electron transport rate, ETR (**C**) in strawberry plants fertilized with BNM medium without phosphorus (Fert. P−) or with BNM medium supplemented with an insoluble phosphorus form (Fert. P+) and subjected to two inoculation treatments after 40, 60, 80, 100, and 120 days of treatment. Values represent mean ± SE of 10 replicates. For statistical details, see Figure 1 caption. (LSD test, * $p < 0.05$, ** $p < 0.01$).

### 3.4. Bacterial Inoculation and Phosphorus Fertilization Availability Effect on Strawberry Ion and C/N Ratio

Leaves and root P concentration and root Ca and K concentrations increased in inoculated plants compared with their non-inoculated counterparts (GLM: leaves and root P,

Inoc., $p < 0.01$; root Ca and K, Inoc., $p < 0.01$; Table 2). This positive effect of inoculation was greater in plants fertilized with additional inorganic phosphorus for root P and K concentrations (GLM: root P and K, Fert., $p < 0.01$; Table 2).

**Table 2.** Leaves and root ion concentrations and C/N ratio in inoculated (+) and non-inoculated (−) strawberry plants with a designed bacterial consortium and fertilized with BNM medium without phosphorus (P−) or with BNM medium supplemented with an insoluble phosphorus form $Ca_3(PO_4)_2$, (P+), after 120 days. Values represent mean ± SE of 10 replicates.

| Treatments | | Leaves Concentrations | | | | | |
|---|---|---|---|---|---|---|---|
| Inoculation | Fertilization | Ca (mg g$^{-1}$) | K (mg g$^{-1}$) | Mg (mg g$^{-1}$) | P (mg g$^{-1}$)$_{Inoc.}$ ** | Mn (mg Kg$^{-1}$) | C/N$_{Inoc., Fert.}$ ** |
| − | P− | 9.4 ± 0.2 | 21.5 ± 0.8 | 5.9 ± 0.2 | 1.2 ± 0.0 | 703.5 ± 12.6 | 27.5 ± 0.1 |
| − | P+ | 9.0 ± 0.1 | 19.6 ± 0.1 | 5.6 ± 0.0 | 1.3 ± 0.0 | 705.1 ± 7.2 | 25.4 ± 0.5 |
| + | P− | 8.9 ± 0.1 | 21.1 ± 0.4 | 5.9 ± 0.1 | 1.6 ± 0.0 | 700.0 ± 10.6 | 24.1 ± 0.2 |
| + | P+ | 8.7 ± 0.1xd | 19.3 ± 0.2 | 5.8 ± 0.1 | 1.7 ± 0.0 | 727.0 ± 6.5 | 22.1 ± 0.1 |
| | | Roots Concentrations | | | | | |
| Inoculation | Fertilization | Ca (mg g$^{-1}$)$_{Inoc.}$ ** | K (mg g$^{-1}$)$_{Inoc., Fert.}$ ** | Mg (mg g$^{-1}$) | P (mg g$^{-1}$)$_{Inoc., Fert.}$ ** | Mn (mg Kg$^{-1}$) | C/N$_{Inoc. x Fert.}$ * |
| − | P− | 4.2 ± 0.1 | 8.9 ± 0.2 | 6.5 ± 0.8 | 1.0 ± 0.1 | 160.0 ± 1.6 | 60.2 ± 0.8 |
| − | P+ | 4.7 ± 0.2 | 10.5 ± 0.9 | 5.3 ± 0.5 | 1.2 ± 0.1 | 160.5 ± 7.1 | 54.3 ± 0.6 |
| + | P− | 5.2 ± 0.2 | 12.9 ± 0.3 | 6.4 ± 0.3 | 1.4 ± 0.0 | 163.3 ± 8.5 | 51.2 ± 1.3 |
| + | P+ | 5.2 ± 0.2 | 14.5 ± 0.7 | 6.4 ± 0.1 | 1.6 ± 0.0 | 180.4 ± 1.0 | 50.8 ± 0.9 |

Fert., Inoc., or Fert. x Inoc. indicate principal or synergistic significant effects (LSD test, * $p < 0.05$, ** $p < 0.01$).

Finally, the tissue C/N ratio decreased in inoculated plants, and overall, it showed a greater decrease in the tissues of plants supplied with inorganic P (GLM: leaves, Fert., $p < 0.01$; root, Inoc. x Fert.; Table 2).

## 4. Discussion

Together with environmental factors such as light, carbon, and water, nutrients are an essential element for plant development. Therefore, a deficient nutrient uptake and/or recycling in soil can often harm plants due to nutrient deficiency [43,44], with this situation being especially serious for agricultural production. This fact has led traditional farmers to apply an excess of fertilizers to ensure the productivity of their crops, with this practice being widely implemented in the strawberry sector [3], with the resulting economic costs and potential environmental risk. As an alternative to chemical fertilization, the value of PGPB as a biotechnology that can increase crop production and improve the efficiency of mineral fertilization, including phosphate, is starting to be recognized [23]. However, this issue has been neglected in the strawberry industry, where most biotechnological advances regarding the use of beneficial microorganisms have focused on the control of pathogens.

A comparative analysis of strawberry plant development subjected to different inoculation and phosphorus fertilization limitation treatments, over 120 days, was able to identify signs of phosphorus deficiency in non-inoculated plants and those fertilized without phosphorous supplementation. Therefore, our results revealed that the leaves of numerous plants had advanced chlorosis and necrosis at the edges, with these being visible signs of an advanced level of phosphorus deficiency [44]. However, biofertilization with the designed PGPB consortium damped the presence of these signs of nutritional stress and, overall, had a positive effect on strawberry plants, in terms of growth, physiological performance, and nutritional balance under phosphorus fertilization withholding. This response would confirm the potential of this biofertilization process to improve the efficiency of strawberry fertilization practices, as has previously been described in other crops, including strawberry [11,45,46].

The positive biofertilization effect in strawberry under the tested limited phosphorus fertilization scenarios could be associated with bacterial consortium PGP properties, which would have had both isolated and combined positive effects on plant performance. In this sense, we should highlight that strains SRT15 and S110, components of the designed consortium, have shown the ability to generate IAA, which is a phytohormone with a key role in its growth (see Table 1). In fact, it is well known that IAA is a key factor in shaping

root architecture during phosphorous starvation [47]. Therefore, the capacity to produce IAA from the strains would directly explain the greater growth of inoculated strawberry, as previously described in strawberry and other horticultural crops [17,48,49], but this effect was mainly seen in plants fertilized with an insoluble phosphorous form. This synergistic response between biofertilization and chemical fertilization treatment could be attributed to the role of the strains SDT3, HPJ40, and SRT15 as phosphate solubilizing bacteria [36,50]. Therefore, these strains would have been able to mobilize insoluble inorganic phosphate due to its metabolic activity, allowing for the formation of bioavailable phosphates for the plant [51,52]. In this way, $Ca_3 (PO_4)_2$ applied during the experiment would constitute a potential source of phosphorus for the plants, buffering its deficit due to bacterial activity, as indicated by the higher leaf and root phosphorus concentration recorded in biofertilized strawberry plants compared to their non-biofertilized counterparts or those inoculated and grown without phosphorous supplementation. Furthermore, the capacity of IAA production of the aforementioned strains would develop a prominent influence in lateral plant root formation, improving its ability to explore a larger soil volume for the acquisition of nutrients [25,26], as is also corroborated by the higher Ca and K concentrations of the roots recorded in this study. Therefore, the long-term effects of biofertilization on strawberry plant growth could be ascribed to the combination of differential development of the plant root system together with the higher bioavailable levels of phosphorus due to the activities IAA and phosphatase of the bacterial strains. This response would generate positive feedback, since a higher root surface would provide a higher plant capacity for nutrient uptake, amplifying the difference in terms of nutritional balance and, consequently, on its development [44,53] between strawberry plants subjected to different combinations of experimental treatments over time.

On the other hand, it should be noted that phosphorus is a structural element of nucleic acids and phospholipids of cell membranes, which is involved in essential metabolism processes such as energy transfer, carbon assimilation, and sugar transformation [54]. In this sense, the greater growth registered in inoculated plants and fertilized with insoluble phosphorus would also have been indirectly related to an improvement in key steps of the plant carbon assimilation and energy transfer efficiency, as indicated by the results of gas exchange and fluorescence analysis. Thus, bacterial inoculation was responsible for ameliorating the deleterious effect of phosphorus deprivation on the carbon absorption capacity of strawberry plants, as evidenced by the higher $A_N$ values obtained in biofertilized plants, and principally in those grown with the supplementation of an insoluble phosphorous form. Therefore, compared to these plants, $A_N$ decreases by 17, 28, and 35% for inoculated plants without phosphorus addition, and in those non-inoculated with and without phosphorous supplementation, respectively. Additionally, our results revealed that the greater $A_N$ rates led to an increase in their water use efficiency, as indicated by the higher values of $_iWUE$ after 80 days of treatment and the consequent higher LWC in inoculated plants.

Our results revealed that the photosynthetic yield variation between the different experimental treatments was mediated by an increase in $CO_2$ diffusion limitation, as indicated by the lowest $g_s$ and $g_m$ values recorded in non-inoculated plants and, to a lesser extent, in those inoculated and grown without phosphorus supplementation. According to our results, some studies have argued that the limitation of P affects photosynthesis through the effect on stomatal functionality, since phosphorus is required for ATP biosynthesis, and is involved in stomatal physiology [55,56], but we also found an important effect on the conductance of the mesophyll to $CO_2$, although this effect has rarely been recorded in the literature [57]. However, it should be highlighted that the quantitative analysis of the absolute photosynthetic limitations also revealed a drastic increase in biochemical limitations in non-inoculated plants, regardless of the fertilization treatment, as reflected by the lower $V_{c,max}$ values together with the increase in $C_i$, despite the decrease in $g_s$. This response could be attributed to a down-regulation of the RuBisCO enzyme, fundamental in the carbon fixation process, which would be partly mediated by the phosphorus

deprivation imposed in this study. In this sense, several studies have denoted the role of phosphorus as a key structural component of fundamental molecules in the photosynthetic pathway, such as RuBisCO and fructose 6-phosphate, meaning that a deficiency of this element would affect the photosynthetic fixation of $CO_2$ [56,58–60] and could contribute to explaining our instantaneous gas exchange results. Similarly, N is one of the most important elements, due to it being a limiting nutrient for plant development and especially photosynthetic metabolism [61]. Thus, a low nitrogen level in plant tissues could lead to several photosynthetic-related enzymes, including RuBisCO, and thus limit photosynthesis metabolism [56]. In this sense, we found a reduction in plant tissues C/N in biofertilized plants, and mainly in those supplied with inorganic phosphorus, which was linked to the higher nitrogen content in its tissues and, therefore, with the nitrogen fixation capacity of the bacterial consortium (i.e., strains SMT38, SRT15, and S110; see Table 1). Therefore, these strains would provide assimilable nitrogen to the plant [62] and, consequently, would contribute to improving the nutritional balance of the plant. Thus, this positive effect together with the aforementioned beneficial uptake of phosphorus, due to the phosphatase activity, contributes to explaining the beneficial impact of biofertilization on strawberry carbon metabolism and, consequently, on plant growth under phosphorus limitation fertilization management.

On the other hand, the beneficial impact of biofertilization on strawberry plants under phosphorus-deprivation conditions was associated with photochemical apparatus responses. Thus, despite the described impact of phosphorus deficit in soil on the PSII photochemistry functionality in several crops species [55,63,64], our results revealed that, overall, $F_v/F_m$ was higher in biofertilized plants, and especially compared which those non-inoculated and grown in the absence of phosphorus supplementation after 80 and 120 days of treatment. This positive response was also corroborated by the invariably higher $\Phi_{PSII}$ and ETR values, which were associated with the maintenance of its electron transport chain functionality. These responses suggest that bacterial inoculation in combination with fertilization with insoluble phosphorus would contribute to improving photosystem efficiency for captured energy absorption, transport, and transformation, providing enough energy (i.e., ATP and NADPH) for $CO_2$ assimilation.

Finally, it should be noted that other consortium strains, such as SMT38 and S110, have the capacity to produce ACC, which is capable of reducing the plant ethylene level [21], helping to overcome the potential growth inhibition induced by environmental stress factors [17,18,50], and therefore, preventing the level of ethylene from becoming growth inhibitory [65], which would allow greater development of the above-ground and below-ground biomasses, which we were able to verify in our experiment.

## 5. Conclusions

Our study is the first to assess the effect of biofertilization, with an isolated PGPB consortium of halophytes integrated by highly multi-stress-resistant bacterial strains and with multiple PGP properties, on strawberry growth, physiological performance, and nutritional balance under phosphorus-limitation fertilization management scenarios. Therefore, we find that bacterial inoculation improves strawberry plant yields under the suboptimal management of phosphorus fertilization. This response is associated with PGPB impact on plants' phosphorus and nitrogen nutritional balance, as denoted by a lower C/N ratio and higher tissue P concentration, which was directly associated with the beneficial impact on carbon absorption capacity and water use efficiency, to counterbalance the limitations of $CO_2$ diffusional and mainly biochemical photosynthesis under conditions of phosphorus fertilization deprivation. In addition, inoculation led to better photochemical apparatus efficiency, in terms of electron transport chain functionality to process greater light for carbon fixation and, consequently, in plant growth under phosphorus-limitation fertilization management. Finally, from an agronomic point of view, this study suggests the possibility of using combined fertilization practices based on biofertilization with PGPB together with a limited application of insoluble phosphorous compounds to achieve more

efficient strawberry P fertilization management practices, providing high cost-effective yield efficiency and decreasing the potentially deleterious environmental impact caused by the overapplication of chemical fertilizers.

**Author Contributions:** E.M.-N. and. S.R.-G. conceived the study, supervised, and acquired funding for the project; E.M.-N. gathered the data and designed and performed the analyses with the help of P.V.-R., J.V.G.-L. and S.R.-G.; N.J.F.-D. and Y.L.I., with the assistance of I.D.R.-L. and E.P., provided the bacterial inocula. All authors performed the experimental development, provided corrections to manuscript drafts, and discussed ideas within it. All authors have read and agreed to the published version of the manuscript.

**Funding:** This work has been supported by the Spanish Government (Ministerio de Ciencia e Innovación-AEI) through the grant project PDC2021-120951-I00 MCIN/AEI/10.13039/501100011033 and UE "Next Generation EU/PRTR".

**Data Availability Statement:** The data presented in this study are available on request from the corresponding author.

**Acknowledgments:** The authors thank the Research General Services of the University of Seville Greenhouse and Herbarium (SGI and SGH, CITIUS) for providing the facilities and equipment.

**Conflicts of Interest:** The authors declare that they have no conflict of interest.

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
