# Peer review of "Biofertilization with PGP Bacteria Improve Strawberry Plant Performance under Sub-Optimum Phosphorus Fertilization"

_agronomy, doi:10.3390/agronomy13020335_

Round 1
Reviewer 1 Report
Comment for the author:
General comment:
§ The uses of bio-fertilizers are now demands of worlds to mitigate the consequence of chemical on soils and animal health. The manuscript is also well laid out. However, the work cannot be accepted in its current state. MS should revise before being publication.
Specific comment
§ MS have too many long phrases. This need to be revised.
§ Mentioned Pot condition, soil texture, chemical composition, amount of water regularly given, optimum humidity, light and dark hour per day, temperature, longitude and latitude positions of places where the experiments was performed, etc.
§ Mentioned the full name of abbreviation, whenever use first time.
§ The software used to calculate the error bar shown in the graphic should be mentioned in MS. It should also be mandatory to indicate whether this is standard deviation or standard error.
§ The bacterial strain name (S110T) in text and table1 (S110) should be identical.
§ The genera name should be italic in MS
Author Response
Dear Editor,
We have carefully revised the manuscript according to the reviewers’ comments and suggestions. We are grateful to reviewers for their comments.
Reviewer #1 comments:
Comment 1: § MS have too many long phrases. This need to be revised. >>All manuscript has been revised in deep and we have reduced the length of some sentences>>
Comment 2: Mentioned Pot condition, soil texture, chemical composition, amount of water regularly given, optimum humidity, light and dark hour per day, temperature, longitude and latitude positions of places where the experiments was performed, etc.>>New technical details have been included in M&M section>>
Reviewer Comment 3: Mentioned the full name of abbreviation, whenever use first time. >>done>>
Reviewer Comment 4: The software used to calculate the error bar shown in the graphic should be mentioned in MS. It should also be mandatory to indicate whether this is standard deviation or standard error. >>This aspect has been clarified in all figure captions. In addition, software for all statistical analysis, included SE, is mentioned in M&M (section 2.6)>>
Reviewer Comment 5: The bacterial strain name (S110T) in text and table1 (S110) should be identical. >>Corrected>>
Reviewer Comment 6: The genera name should be italic in MS. >>Checked and corrected>>
Reviewer 2 Report
This article presented Biofertilization with PGP Bacteria improve Strawberry Plant Performance under Sub-optimum Phosphorus Fertilization. However, there are some shortcomings for that should be revised.
The abstract is very general written. No specific or quantitative results are provided.
The authors should also present specific methods in the abstract.
The authors should provide economic and industrial importance of the strawberry in introduction.
Line 55 should be cite with relevant study. The following study would be helpful.
https://doi.org/10.1002/aoc.5190,
Line 64-65 could be cited with relevant study. 10.1016/j.micpath.2020.103966
The authors should provide complete details that how the samples were sterilized. Which specific environment was provided under green house.
How the it was make sure that the growth was only effected by PGPBs
How bacterial strain were identified.
Provide details of the structure and shape of isolated bacterial strains.
Which media was provided for bacterial strains incubation and preservation.
Which statistical analysis was applied on table 2.
Conclusion is well presented. However, future recommendations based on the obtained results must be added in the conclusion section.
Author Response
Dear Editor,
We have carefully revised the manuscript according to the reviewers’ comments and suggestions. We are grateful to reviewers for their comments.
Reviewer #2 comments:
Comment 1: The abstract is very general written. No specific or quantitative results are provided. The authors should also present specific methods in the abstract. >>M&M details and quantitative results have been included in abstract section>>
Comment 2:The authors should provide economic and industrial importance of the strawberry in introduction. >>worldwide strawberry production, has been included in Introduction section, following reviewer recommendation>>
Comment 3:Line 55 should be cite with relevant study. The following study would be helpful. https://doi.org/10.1002/aoc.5190 >>included>>
Comment 4: Line 64-65 could be cited with relevant study. 10.1016/j.micpath.2020.103966 >>included>>
Comment 5:The authors should provide complete details that how the samples were sterilized. Which specific environment was provided under greenhouse. >>These M&M details have been included>>
Comment 6:How the it was make sure that the growth was only effected by PGPBs >>The inoculations were carried out in a sterile soil, always using a control without inoculation to be able to observe the effect of the inoculations. With this procedure the effect of the PGPBs on the plants can be checked>>.
Comment 7:How bacterial strain were identified. >>Bacteria were separated based on the different colony morphology and cell morphology was observed by Gram stain using an Olympus CX41 microscope with the 100x objective and identified by 16S rRNA gene amplification using 16F27 and 16R1488 primers. This methodology is described in deep in our previous studies. Please see references [31-36] >>
Comment 8: Provide details of the structure and shape of isolated bacterial strains. >>structure and shape of the isolated bacterial strains have been previously described in our previous studies, please see Flores-Duarte et al., 2022 (Variovorax paradoxus, S110: Gram-negative bacilli); Andrades-Moreno et al., 2014, Pseudomonas sp. SDT3: Gram-negative bacilli); Mesa et al., 2015 (Bacillus velezensis, SMT38: Gram-positive bacilli); Mesa-Marín et al., 2020 (Pseudarthrobacter oxydans, SRT15: Gram-positive bacilli); Redondo-Gómez et al., 2022 (Bacillus zhangzhouensis, HPJ40: Gram-positive bacilli). >>
Comment 9: Which media was provided for bacterial strains incubation and preservation. >>The strains were cultivated in TSA medium (Tryptone Soya Agar, Intron Biotechnology, Gyeonggi-do, Korea) for 24 hours at 28â—¦C. Different bacteria were separated in based on the color and morphology of colonies and Gram stain for 24 hours at 28â—¦C. Pure cultures were stored in sterile containers with tryptic soy broth (TSB) medium (Intron Biotechnology, Gyeonggi-do, Korea) and 15% (v/v) glycerol at -80â—¦C. These technical details have been described in deep in our previous studies. Please see references [31-36].
This information has not been included in the study as it is information of a technical nature, already published.>>
Comment 10: Which statistical analysis was applied on table 2. >>This information is included in the bottom part of the table 2>>
Comment 11: Conclusion is well presented. However, future recommendations based on the obtained results must be added in the conclusion section. >>Agronomic recommendation based on the obtained results are included at the end the conclusion>>